# Three-Dimensional Chromatin Structure of the EBV Genome: A Crucial Factor in Viral Infection

**DOI:** 10.3390/v15051088

**Published:** 2023-04-29

**Authors:** Lisa Beatrice Caruso, Davide Maestri, Italo Tempera

**Affiliations:** 1The Wistar Institute, Philadelphia, PA 19104, USA; lcaruso@wistar.org (L.B.C.); dmaestri@wistar.org (D.M.); 2Department of Pharmacy and Biotechnology, University of Bologna, 40126 Bologna, Italy

**Keywords:** EBV, latency, chromatin structure, chromatin looping, epigenetics, CTCF, cohesin

## Abstract

Epstein–Barr Virus (EBV) is a human gamma-herpesvirus that is widespread worldwide. To this day, about 200,000 cancer cases per year are attributed to EBV infection. EBV is capable of infecting both B cells and epithelial cells. Upon entry, viral DNA reaches the nucleus and undergoes a process of circularization and chromatinization and establishes a latent lifelong infection in host cells. There are different types of latency all characterized by different expressions of latent viral genes correlated with a different three-dimensional architecture of the viral genome. There are multiple factors involved in the regulation and maintenance of this three-dimensional organization, such as CTCF, PARP1, MYC and Nuclear Lamina, emphasizing its central role in latency maintenance.

## 1. Introduction

Epstein–Barr Virus, or EBV, is a human gammaherpesvirus that infects 95% of the population worldwide [1,2,3]. EBV infects both epithelial and B cells and subsequently establishes a lifelong persistent infection in B cells, where it remains as a chromatinized episome with multiple copies per cell [4]. Usually, EBV persistent infection is asymptomatic, yet nearly 140,000 people die annually from untreatable malignancies caused by EBV infection of lymphoid or epithelial cells [5,6]. Indeed, EBV infection is causally associated with several malignancies, including post-transplant lymphoproliferative disorders (PTLD) [7], Burkitt’s Lymphoma (BL) [1], Diffuse Large B-cell Lymphoma (DLBCL), gastric carcinomas (GC), and nasopharyngeal carcinoma (NPC) [8,9,10]. In EBV-associated malignancies, viral gene expression is limited to a few viral genes that encode for viral proteins without producing viral particles [11,12,13]. This type of infection is referred to as latent infection. Latent infection represents an oncogenic force in establishing and maintaining the transformed phenotype of infected cells, although seminal studies have highlighted the importance of lytic reactivation, or abortive lytic reactivation, as a factor in transformation [14,15,16,17].

In latently infected cells, EBV expresses a limited set of viral proteins: six Epstein–Barr Nuclear Antigens (EBNAs); three integral membrane proteins called Latency Membrane Protein (LMP1 and LMP2a and b); two small non-polyadenylated RNAs (EBERs); and several miRNAs generated from the BamHI-A Rightwards Transcripts (BARTs) (Figure 1). EBV expresses these latent viral transcripts in different transcriptional programs, which are referred to as latency types [18,19,20]. The different latency types observed in vitro, and characteristic of each EBV-associated tumor, mirror transcriptional changes occurring in viral gene expression while EBV is transitioning in either the lymphoid or epithelial compartment [21,22,23]. Thus, during latency, EBV gene expression is quite dynamic, and the switching between different latency programs is a critical event for the establishment and maintenance of EBV latency and for triggering and sustaining the proliferation of either B or epithelial cells [24,25].

## 2. EBV Latency

EBV can adopt at least three different latency types in latently infected cells (Figure 2). In Type I latency, EBV expresses only the non-coding RNAs EBERs and EBNA1, the transcription of which is initiated from the Qp viral promoter [26,27,28]. Interestingly, the 3′ region to the Transcription Start Site (TSS) of the Qp promoter contains two EBNA1-binding sites which, when bound by EBNA1, repress Qp activity, providing a self-regulating mechanism to control EBNA1 expression in Type I latency [27,29,30]. Type I latency is observed in EBV+ BL tumors, in BL cell lines and in memory B cells from healthy individuals. In Type II latency, EBV expresses the two LMPs, the non-coding RNAs EBERs and BARTs in addition to EBNA1 from the Qp promoter. Type II latency is observed in vivo in nasopharyngeal carcinoma and Hodgkin disease (HD) cells [31]. However, in the B cells from patients with EBV Infectious Mononucleosis (IM), it has been observed that the EBV expresses the EBERs and the EBNAs at high levels while the LMPs genes are expressed at low levels or not expressed at all. This type of latency with partiality has been referred as Type IIb latency [32]. In Type III latency, EBV expresses all the latent viral transcripts, consisting of the six EBNAs, the two LMPs, and the EBER and BART non-coding RNAs [33]. The transcription of the EBNAs, including EBNA-1, is initiated by the Cp promoter during Type III latency [23]. The Cp promoter contains an EBNA-2 responsive element upstream of the 5’ of TSS, and the binding of EBNA-2 and EBNA-LP upregulates Cp activity, leading to positive autoregulation of EBNA transcripts [26,34,35,36,37]. Type III latency is observed in vitro in proliferating primary B cells after EBV infection and in vivo in PTLD and DLBC lymphoma cells [23,32,38,39].

Epigenetic regulation of EBV latency programs. Changes in the viral promoter that the virus uses determine which latency state the EBV-infected B cells adopt [26,40]. EBNA-2, EBNA-LP and EBNA-1 viral proteins contribute to regulating the activity of Cp and Qp, respectively. Moreover, EBNA-2 and EBNA-LP contribute to upregulating the LMP promoter’s activity [36,37]. However, cellular factors, including epigenetic regulators, control the two promoters [20,41,42]. The importance of epigenetic modifications for regulating EBV gene expression emerged from early studies, showing that treating EBV+ B cells with hypomethylating agents induced EBV viral replication [43,44,45,46].

Further studies determined that the EBV genome was highly methylated, restricting viral genes expression in both B and epithelial cells [46]. Subsequent studies also determined that treatment of EBV-infected cells with inhibitors of histone deacetylases induced the reactivation of EBV replication, indicating that DNA methylation and heterochromatinization of the viral genome represent an important repressive mechanism for viral replication [47,48]. Interestingly, analysis of CpG methylation across the EBV genome during in vitro infection of primary B cells demonstrated that methylation of the viral genomes is a slow process that requires several weeks post-infection for completion, suggesting that other epigenetic and cellular factors may be fundamental to the early regulation of viral gene expression [49,50,51,52]. In contrast, DNA methylation may be essential in controlling and maintaining viral gene expression later during EBV infection. Consistent with these observations, the extent and the distribution of DNA methylation across the EBV genome differs between latency types, with high levels of methylation observed in Type I infected cells, indicating that DNA methylation is an essential epigenetic mechanism for maintaining latency programs [53,54,55].

The notion that each latency type is characterized by a specific epigenetic landscape across the EBV genome is further supported by early studies demonstrating that the deposition of different patterns of histone modifications correlates with different EBV latency types [53,56,57]. These studies showed that in Type III latency, the most permissive type of latency concerning latent viral genes expressed, the EBV genome was enriched with histone marks associated with open chromatin and active gene expression, including H3K27ac and H3K4me3 [53,56]. In Type I latency, where the EBV gene expression is limited to only EBNA-1, the viral epigenetic landscape is characterized by the deposition of repressive histone marks such as H3K9me3 and H3K27me3 [53,56], the latter being deposited on the chromatin by EZH2, the catalytic subunit of the Polycomb Repressive Complex 2 [58]. Interestingly, during latency EZH2 complex binds the EBV genome at the promoter region of lytic genes, but upon reactivation of EBV, the association of EZH2 with the viral genome is lost, and H3K27me3 mark is erased around lytic genes [59,60,61]. Moreover, recent studies revealed that the TRIM28/KRAB-ZFP/SZF1 complex plays an essential role in repressing EBV lytic gene expression by promoting the deposition of the heterochromatin mark at the promoter of lytic genes, facilitating the establishment of EBV latent infection [62,63,64,65]. Thus, the stable EBV gene expression patterns observed during latency depend on histone deposition and DNA methylation across the viral genome.

## 3. EBV and CTCF

Systemic mapping of chromatin has shown that the epigenome is organized into distinct domains of transcriptionally active and inactive regions [56]. These distinct domains are maintained by protein factors that prevent the spread of one domain to the next. CTCF (CCCTC-binding factor) is a highly conserved zinc finger protein that plays an essential role in chromatin organization and gene regulation [66,67,68]. CTCF plays a critical role in organizing chromatin domains by binding to specific DNA sequences known as insulator elements which prevent the spread of epigenetic modifications and maintain the integrity of gene expression programs [68].

In recent years, several studies showed the role of CTCF in the context of EBV infection. CTCF has been shown to bind to the EBV genome during latency [53,55,69,70]. Extensive mapping of CTCF occupancy identified at least 17 CTCF binding sites across the EBV genome [24,56,57,71,72]. In particular, CTCF has been shown to bind to the latent promoters Cp, Qp and LMPs, as well as the promoter of the BZLF1 gene, which encodes for the lytic transactivator Zta [56]. Surprisingly, no differences in CTCF binding across the viral genome were found between latency types [24], although between Type III and Type I latency, a difference in the CTCF binding strength at Cp promoter was observed [69]. However, EBV genomes that carry mutations disrupting the CTCF binding either at Cp, Qp, or LMPs show impaired activity of those promoters and altered chromatin composition of the neighboring regions [53,69,73]. In particular, the disruption of CTCF binding at the Qp promoter in Type I latently infected epithelial cells resulted in the spread of H3K9me3 repressive heterochromatin mark and the accumulation of DNA methylation at the Qp region over time, leading to promoter silencing and inhibition of EBNA-1 expression [53]. Accordingly, it can be suggested that CTCF binding across the EBV genome physically acts as a barrier that prevents the unregulated spreading of epigenetic modifications into viral promoter regions, maintaining the integrity of latency gene expression programs. Contrary to what was observed for the latent viral promoter, disruption of CTCF binding to the BZLF1 promoter failed to reactivate lytic infection of EBV and no significant changes in CTCF binding across the viral genome were observed during the early phase of EBV reactivation, suggesting that CTCF binding per se is not sufficient to completely reverse the epigenetic silencing of lytic promoters [72].

Furthermore, several studies have revealed that CTCF can regulate viral gene expression in other double-stranded DNA viruses. For instance, in Kaposi’s sarcoma herpesvirus (KSHV) [74,75,76,77,78,79,80,81,82], herpes simplex virus (HSV) [83,84,85,86,87,88,89] and human cytomegalovirus (CMV) [90], CTCF has been shown to bind to the viral genome and regulate viral gene expression by modulating the accessibility of the viral genome to transcription factors. Finally, it has also been suggested that viruses can modulate CTCF activity to promote viral replication. For example, it has been proven that human papillomavirus (HPV) can induce changes in CTCF binding that promote viral replication and transcription [91,92,93]. Overall, the role of CTCF in the context of viral infections is an area of active research, and further studies are needed to fully understand the mechanisms by which CTCF regulates viral gene expression and how viruses modulate CTCF activity.

## 4. CTCF and Cohesin

CTCF also plays a critical role in regulating gene expression by influencing the three-dimensional structure of chromatin and promoting or inhibiting interactions between enhancer and promoter gene regions [68]. Recently, it has been discovered that CTCF often works together with Cohesin to regulate gene expression and chromosome architecture [94,95,96]. Cohesin also plays a role in regulating gene expression by promoting the formation of chromatin loops and facilitating interactions between regulatory elements. Cohesin is a protein complex critical to chromosome segregation during cell division [97]. The cohesin complex is responsible for holding sister chromatids together and then releasing them at the onset of mitosis [98]. Cohesin can bind to CTCF at specific sites to form chromatin loops that bring enhancers and promoters into proximity, leading to increased gene expression [94,99]. In addition, CTCF binding can act as a barrier to prevent Cohesin from spreading along the chromatin fiber, limiting its effects to specific regions [100]. Overall, CTCF and Cohesin are critical components of the regulatory network that governs gene expression and chromosome architecture in eukaryotic cells. Consistent with these observations, in latently infected cells, extensive mapping of EBV epigenome revealed that Cohesin occupancy overlaps with CTCF binding to specific regions of the viral genome, including the Cp, Qp, BZLF1, and LMP1 promoters [24,56,57].

The role of CTCF and Cohesin in regulating the chromatin architecture of the genome in higher eukaryotes prompted similar studies to determine the 3D structure of the EBV genome in latently infected cells. Earlier studies focusing on the 3D structure of the Cp and Qp regions of the EBV genome demonstrated that these regions adopt alternative 3D chromatin structures between latency types [25] (Figure 1). For example, in Type III latency, the active Cp promoter forms a chromatin loop with the Ori P region, the origin of DNA viral replication during latency, which also serves as a transcriptional enhancer [25,101]. On the contrary, in Type I latency, where Cp is repressed and transcription of EBNA-1 is initiated from the Qp promoter, a chromatin loop between Qp and Ori P was observed [25]. In addition, a chromatin loop that brings Ori P close to the LMP1 promoter was observed in Type III latency, indicating that chromatin loop formation is implicated in regulating viral gene expression during EBV latency [56,73]. All these chromatin loops connecting Ori P with the viral promoters Cp, Qp, and LMPs occur at regions of the EBV genome where CTCF and Cohesin bind, indicating that CTCF and Cohesin actively participate in the formation of chromatin loops across the viral genome. Indeed, in EBV genome carrying mutations that ablate CTCF and Cohesin binding at either Cp, Qp, or LMP promoters, no chromatin loops occur between these regions and Ori P [25,73], indicating that CTCF and Cohesin binding is essential for chromatin loop formation between viral genomic regions.

Most recently, studies employing EBV-specific Capture-HiC assay revealed the chromatin architecture of the EBV genome in Type I and Type III EBV+ B cells (Figure 3) [24]. These studies showed several chromatin loops across the viral genome, connecting regulatory DNA elements to viral promoters. Notably, chromatin loops occur between viral regions that contain at least one CTCF binding site [24]. In Type III cells, more EBV genomic regions are connected through chromatin loops than in Type I cells, in which instead, very few prolonged distant interactions occur, indicating that the frequency of chromatin loop and complexity of 3D structure in EBV latency correlates to the level of transcriptional permissiveness of latent viral genome [24]. However, several viral regions are engaged in similar chromatin loops in both Type I and Type III EBV+ cells. For example, the region upstream of the LMP2 promoter is connected to the regions encoding for the EBERs in both latency types. Similarly, in both Type I and Type III EBV+ cells, a chromatin loop connects the origin of lytic replication OriLyt Left to the CTCF site positioned at the 3′ of W repeats, suggesting a potential role of 3D structure in restricting lytic reactivation [24]. Consistent with a possible role for chromatin looping in controlling lytic replication, work from the Gewurz group (discussed later in this manuscript) recently demonstrated that OriLyt Left connects to the Zp promoter through a chromatin loop upon lytic reactivation [102].

## 5. Regulation of EBV 3D Structure

Between latency types, the EBV genome can therefore assume alternative 3D structures, indicating that the chromatin architecture provides an additional layer of epigenetic regulation for EBV gene expression during latency. Genetic studies show that CTCF is essential for forming chromatin loops, and EBV chromatin loops occur between viral regions occupied by CTCF. Yet, the CTCF binding profile across the EBV genome is similar between Type I and Type III EBV+ cells [24]. Most recently, this discrepancy between similar CTCF binding and different 3D chromatin structures in Type I and Type III EBV+ cells has been attributed, at least in part, to the effect of Poly (ADP-ribose)polymerase 1 (PARP1) on CTCF [24,103]. PARP1 is a protein that catalyzes the apposition of ADP ribose polymers to acceptor proteins, including histones and CTCF [104,105,106]. PARP1 physically interacts with CTCF, and PARylation of CTCF facilitates its functions, including chromatin loop formation [105]. EBV infection can activate PARP1, in part through the signaling cascade initiated by LMP1 [107]. PARP1 binds to CTCF at specific regions of the EBV genome, and pharmacological inhibition of PARP1 destabilizes CTCF binding to some (but not all) regions of the EBV genome [103]. In latency III EBV+ B cells, the inhibition of PARP1 ablates viral chromatin architecture, causing heterochromatinization of the viral episome and repression of EBV viral genes [24,103] (Figure 3). For example, in Type III EBV+ cells, PARP inhibition significantly decreases CTCF occupancy at the Cp promoter and alters the 3D chromatin structure of this promoter region, reducing the expression of EBNA2 [24,103]. However, it is worth noting that the effect of PARP1 inhibition is limited to only a subset of chromatin loops present across the EBV genome, indicating that other mechanisms, besides PARP1 activity, regulate the organization of the EBV tridimensional chromatin structure. Interestingly, PARP1 plays a role in the infection of other DNA viruses, including HSV-1 [108,109] and KSHV [110,111,112], which are also epigenetically regulated by CTCF, thus suggesting that PARP1 and CTCF interaction might be a common regulatory axis of viral infection.

## 6. EBV and Nuclear Lamina

In eukaryotes, an additional level of epigenetic regulation that can influence the chromatin structure is the interaction of genomic regions with the Nuclear Lamina (NL). The Nuclear Lamina is formed by proteins named lamins. They are grouped into two different families (type A and B lamins) that are located at the nuclear peripheral space. The nuclear lamina–genome interactions occur at chromatin regions called Lamin-Associated Domains (LADs) that when located at the nuclear periphery are associated with different repressive histone marks, such as H3K9me2/3 and H3K27me3 [113,114,115], and have low transcription levels.

In the context of herpesvirus biology, the nuclear lamina has been reported to play many different roles in regulating different stages of viral infection. To be specific, the nuclear lamina can control viral replication by acting as a physical barrier that prevents the viral capsid egress by inducing the expression of lytic genes that target the nuclear lamina itself. For example, EBV lytic gene BGLF4 encodes a viral kinase that phosphorylates lamina A causing its degradation [116]. Alternatively, the nuclear lamina can also be involved in the formation of viral replication compartments [117,118,119,120,121]. EBV latent infection of B cells induces Lamin A/C expression, and Type III but not Type I EBV+ B cells express Lamin A/C [122].

The nuclear lamina has been also reported to play a role in EBV latency [122]. LADs were found across the EBV genome, with differences in binding patterns for both lamin B1 and lamin A/C depending on the EBV latency types. To be specific, in Type I, a prevalence in lamin B1 binding to the EBV genome was found, while in Type III, lamin A/C was the one mostly binding to the EBV genome [122]. These differences in binding profile correlate with differences in the transcriptional profile in the two latent programs. Remarkably, in EBV+ gastric cancer cells, in which viral gene expression is limited, genome-wide associations between the EBV and the host genomes tend to occur at DNA regions associated with the nuclear lamina [123].

Lamin B1 binding is conserved in both latency types at both the origins of lytic replication OriLyt Left and RMPS1-OriLyt, which are regions inactive during latency [122], indicating an ulterior way by which the nuclear lamina can regulate the viral lytic replication. Consistently, it was found that during EBV lytic reactivation, when both OriLyt regions are active, the interactions between EBV and human LADs decrease [124].

The expression of Lamin A/C was also involved in the modulation of expression of both latent and lytic genes, suggesting a role of lamin A/C in fine tuning EBV gene expression during latency [122].

Furthermore, the nuclear lamina–genome interactions regulate gene expression by modifying the chromatin composition without affecting its 3D structure. In Type III latency, the lamin A/C depletion resulted in reduced H3K9me2 deposition at viral loci bound by both lamin A/C and lamin B1 [122]. The H3K9me2 histone mark associates with heterochromatin regions localized at the nuclear periphery, and reduced levels of this mark at LADs impair the association of these regions with the nuclear lamina [125,126].

## 7. EBV and MYC

Besides the central role of CTCF/Cohesin complex in regulating the 3D structure of EBV, another transcription factor involved in this process is Myc. The Myc protooncogenes are upregulated in a wide variety of cancers and their expression is induced upon EBV infection [127]. The work mentioned [102] has demonstrated how its levels are used by EBV as a sensor of the B cell state and its binding to the viral genome helps maintain latency by restricting the expression of lytic genes.

Depletion of Myc promotes the reactivation of the lytic cycle through the formation of loops between the promoter of the early lytic gene BZLF1 and the OriLyt enhancer and Terminal Repeats (TR) regions.

## 8. EBV-Associated Enhancers

As discussed above, Cohesin and CTCF are essential to the regulation of gene expression, placing distal enhancers and promoters in close proximity. The enhancers are enriched in two histone modifications, H3K4me1 and H3K27ac, which, respectively, characterize poised and active enhancers. Extensive mapping of the H3K27ac histone mark revealed that the EBV genome is enriched for this histone mark at key latent promoters such as Ori P, Cp, Qp, LMP and some newly discovered regions such as the BILF2 promoter [128]. In accordance with the central role of CTCF in the 3D organization of the viral genome and its role in regulating viral gene expression, all H3K27ac-enriched regions are near CTCF binding sites.

With newly developed techniques such as HiChIP that combine chromatin immunoprecipitation with HiC, it has been possible to map the intragenomic interactions that are enriched for H3K27ac in B cells harboring all three latency types [128]. In this study, Type I latency genomes showed few strong enhancers and enhancer–promoter interactions, whereas Type II and III latencies exhibited multiple strong enhancers throughout the viral genome.

## 9. Conclusions

The three-dimensional structure of EBV is closely associated with the latency program and, therefore, with the regulation of viral gene expression. The central role of viral chromatin architecture is underscored by the fact that multiple factors are exploited by the virus to maintain it, including CTCF, Cohesin, PARP1, MYC, and Lamins. However, our present understanding of the role of chromatin structure in EBV gene expression is still incomplete due to technological limitations. For example, multiple copies of EBV episomes exist in each infected cell, yet HiC technologies only provide the viral 3D genome structure obtained from a cell population. Therefore, it is unclear whether all EBV episomes within an infected cell adopt the same chromatin structure or whether sub-populations of episomes exist, each adopting a specific 3D chromatin structure consisting of just a sub-set of loops. Overall, shedding light on these regulatory layers can be of considerable importance in the treatment of EBV-associated malignances.

In recent years, numerous drugs targeting epigenetic mechanisms have been developed and utilized to improve the efficacy of chemotherapy. Considering the significant role of epigenetics in regulating both EBV infection and the development of EBV-associated malignancies, leveraging epigenetic drugs may offer a promising and innovative therapeutic strategy for treating EBV-positive malignancies.

## Figures and Tables

**Figure 1 viruses-15-01088-f001:**
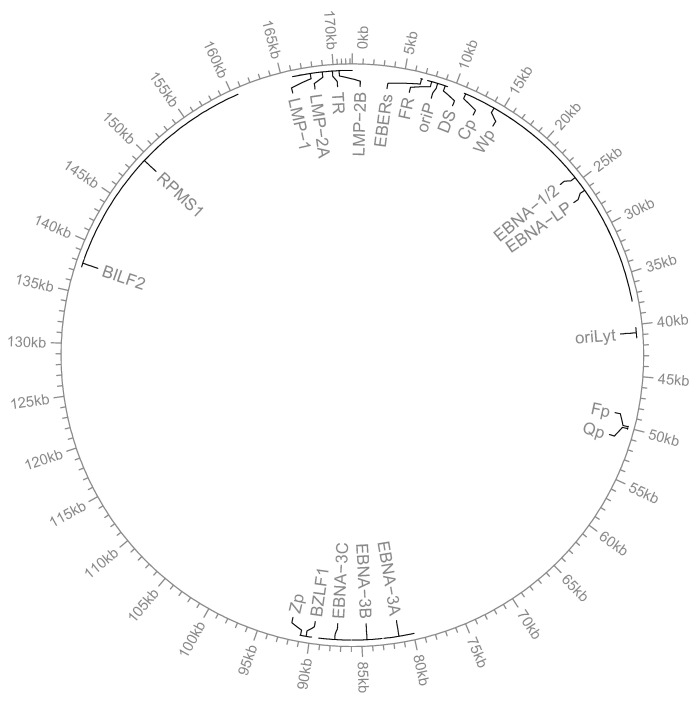
Circular view of the EBV genome. The EBV genome is represented as a circle. For better visualization, only the viral genes and promoters that will be mentioned and discussed in this review are listed.

**Figure 2 viruses-15-01088-f002:**
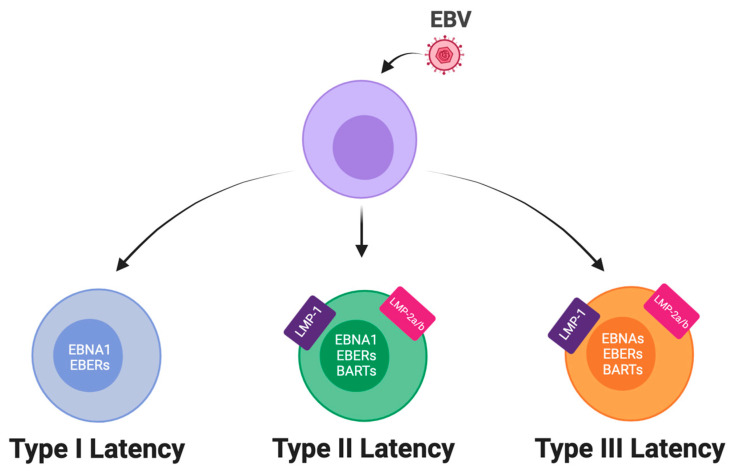
Schematic view of the different latency types. Upon infection, EBV is able to establish latent infection in the host cell. The three main types of latency are depicted in this image. From the left: Type I latency is characterized by the expression of only the viral nuclear protein EBNA1 and the noncoding microRNAs EBERs; Type II latency involves the expression of EBNA1, EBERs, BARTs and the three transmembrane proteins LMP1, 2a, 2b. Type III latency is characterized by the expression of all the previously mentioned genes and the viral transcription factors EBNA2, EBNA3A, EBNA3B, EBNA3C, and EBNA-LP. Created with BioRender.com (accessed on 2 April 2023).

**Figure 3 viruses-15-01088-f003:**
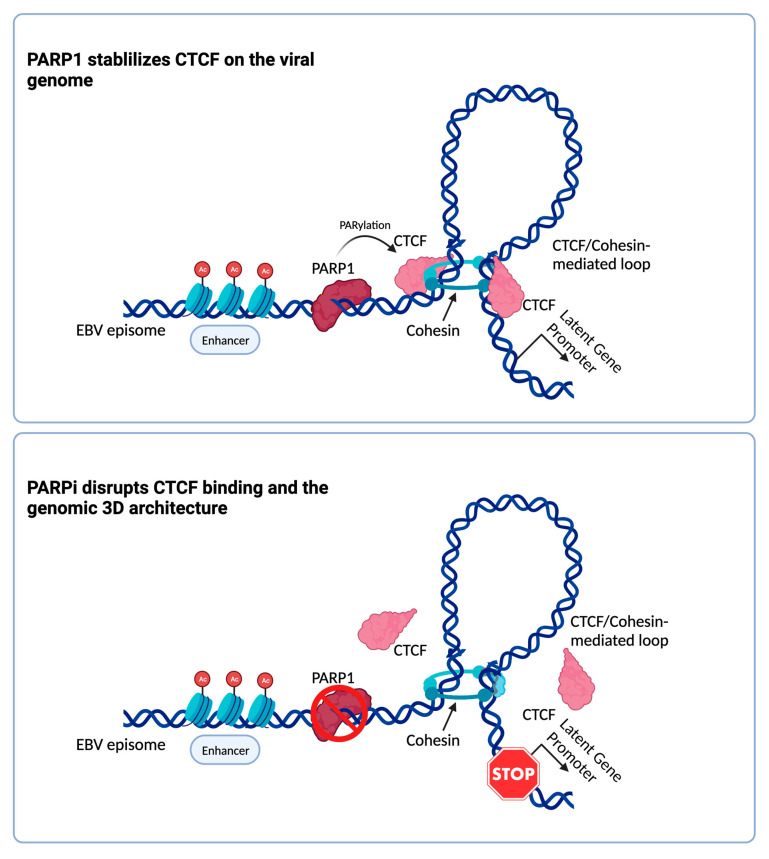
Schematic view on the 3D structure of the EBV genome. (**Top**) PARP1 PARylates CTCF, thus stabilizing its binding on the viral genome. CTCF, together with Cohesin, forms loops in close proximity with viral promoters with enhancers, promoting viral latent gene expression. (**Bottom**) PARP1 inhibition determines a reduction in CTCF binding, therefore causing a disruption of loops and a consequent decrease in viral gene expression. Created with BioRender.com (accessed on 2 April 2023).

## Data Availability

Not applicable.

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
