# Peer review of "Three-Dimensional Chromatin Structure of the EBV Genome: A Crucial Factor in Viral Infection"

_viruses, 2023, doi:10.3390/v15051088_

Round 1

Reviewer 1 Report

This paper summarizes our current knowledge on the regulatory mechanisms of EBV gene expression that governs the status of infection: lytic cycle and multiple types of latency. I find the review quite informative, and yet I have several questions/suggestions that I would like the authors to consider before publication.

Major points

1.     This manuscript lacks Abstract.

2.     The meaning of the term “3D structure” used in this paper may not be common for the general reader. For example, biochemists would have an impression that the authors are talking about the structure of molecules or molecular complexes that can be solved by X-ray crystallography, NMR, etc., and they might be disappointed to learn that is not the case. The term might be already around in this particular research field, but I would suggest the authors to consider using an alternative term to avoid confusion at this point. To me, term like secondary structure or chromatin conformation sounds more appropriate.   

3.     Some abbreviations are used without definition: e.g., TSS and OriLyt Left. I would suggest the authors to define them in the text and to include a list of all abbreviations and their full spelling (and short descriptions in the cases of relatively uncommon terms) on the first page.

4.     It would be helpful for the general reader if the authors include two additional display items: one illustrating the locations of the critical elements and genes discussed in this paper on the EBV genome, and the other one describing the types of latency with genes being expressed. Although the later is described in the text (the section “EBV latency”), the concepts are repeatedly mentioned in the following parts, and it is hard for the general reader to remember each latency type.

Minor points

5.     It is unclear what the term “chromatized” (line 13) means.

6.     In line 24, the authors state “two integral membrane proteins”, but three proteins seem to be listed in the following parenthesis. Besides I guess the “Latency Membrane Protein” should be in plural form. 

7.     In line 36, the phrase “the 3’ --- contains” sounds awkward. I would put it like “the region 3’ --- contains”.

8.     The usage of punctuations (especially comma) is inconsistent or uncommon. For instance, when listing more than three items, the authors put a comma before “and” (i.e., “A, B, and C”) in some cases but not always: e.g., line 38 (after “cell line”), line 40 (after “EBERs”), line 111 (after “[83-89]”), and line 243 (after “LMP”). They should be consistent. Also, I find some punctuations unnecessary: e.g., two commas in line 53, one in line 56 and in line 96, the second comma in line 102 and in line 114, and a semicolon at the end of line 240. On the other hand, necessary commas are missing at some places: e.g., line 53 (after “latency”), line 195 (after “eukaryotes”), line 200 (after “[113-115]”), line 208 (after “expression”), line 212 (after Type I), and line 219 (after “consistently”). 

9.     Usage of “or” is probably inappropriate (“and” is better) at some places: e.g., line 41 and line 47.

10.  The authors tend to use multiple gerunds in a sentence, which sometimes makes sentences hard to understand. For instance, it was impossible for me to decipher the sentence “In specific, ---- the nuclear lamina” (lines 203-206). 

11.  The expressions like “studies have investigated” (line 90), “mapping found” (line 131), and “studies mapped” (line 157) are logically incorrect and unacceptable. Note that it is researchers, not studies (or mapping), that investigate (or find or map) something. In the second case (line 131), other words like “revealed” is acceptable. 

12.  Line 78, “epigenetically repress”: The phrase “play a role in” should be followed by a noun phrase (e.g., “epigenetic repression of”).  

13.  Line 185, “heterochromatinization of the viral epigenome”: Is this expression common in this field?  To me, “genome” instead of “epigenome” sounds more appropriate. 

14.  Line 195, “epigenome regulation”: Similarly, “epigenetic regulation” sounds better to me, unless the authors mean something else. 

15.  Line 196, “nuclear proteins named Nuclear Lamina (NL)”: I believe that NL is the name of subcellular structure, not proteins. I would use “the structure called” or simply omit “nuclear proteins named”. 

16.  Line 201, “transcriptional levels”: This phrase sounds misleading; “transcription levels” should be better.

17.  Line 202, “herpesviruses biology”: I would use the singular form “herpesvirus” to make it adjective.  

18.  Line 203, “the herpesvirus’s”: Is this phrase necessary?

19.  Line 212, “instead”: Is this word necessary? I would use “while” after the comma.   

20.  Lines 221-222, “The expression --- latency”: Papers supporting this statement should be cited (i.e., references are missing). 

21.  Line 231, “As previously mentioned, --- have demonstrated”: This phrase sounds like the finding has been mentioned earlier in this paper (or in some previous paper by the authors). The work, but not the finding, is mentioned earlier (line 169). In this case, I would just say “The work mentioned early [122] has demonstrated”.

22.  Line 239, “before”: This sounds like talking about a previous paper. I would use “above” in this context.

23.  Line 242, “at the level of”: It is unclear what this phrase means in this context.

24.  Line 249, “whether”: Is this correct? To me, “whereas” sounds more appropriate.  

25.  Lines 254-255, “Its central role --- Lamin”: This sentence has several problems. First, it is unclear what “Its” means. Second, it is unclear what the phrase “such as --- Lamin” explains. I guess this phrase lists the examples of “multiple factors”. Third, if so, “regulatory axis” is not a factor. I would therefore suggest the authors to rephrase this sentence.   

I am not a native speaker, but I found several problems in the text and listed them in my review report as Minor points. 

Author Response

We thank the reviewer for the comments provided. In the revised version of the manuscript, we did our best to follow the reviewer's suggestions as follows:

Major:

  1. We added the abstract
  2. Where it was possible, we used the term "secondary structure of the chromatin" in the text. However, it is worth noticing that in the field of epigenetics, terms such as "3D structure" and "chromatin structure" are standard and used in the same way we used them in this review. 
  3. Whenever it was possible, we better defined the definition of abbreviations used in the text of this revised manuscript.
  4. In the revised manuscript, we included two new figures: one illustrating the locations of the critical elements and genes discussed in this paper on the EBV genome. The other describes the types of latency with genes being expressed.

Minor points: we addressed those points in the text of the revised manuscript/

Reviewer 2 Report

The authors nicely summarized the current status of 3D genome structure of EBV genome research.

Minor point: the authors should add a subtitle for the top and bottom panels of the figure.

Author Response

We thank the reviewer for the comments provided.

Minor point:

In the revised manuscript, we added a subtitle for the top and bottom panels of the figure indicated.